# Short-PHD: Detecting Short LLM-generated Text with Topological Data Analysis After Off-topic Content Insertion

**Dongjun Wei[1], Minjia Mao[2], Xiao Fang[2] & Michael Chau[1]***
[1] The University of Hong Kong, Hong Kong, China
[2] University of Delaware, DE 19716, USA
dongjun@connect.hku.hk, {mjmao,xfang}@udel.edu, mchau@business.hku.hk

## Abstract

The malicious usage of large language models (LLMs) has motivated the detection of LLM-generated texts. Previous work in topological data analysis shows that the persistent homology dimension (PHD) of text embeddings can serve as a more robust and promising score than other zero-shot methods. However, effectively detecting short LLM-generated texts remains a challenge. This paper presents Short-PHD, a zero-shot LLM-generated text detection method tailored for short texts. Short-PHD stabilizes the estimation of the previous PHD method for short texts by inserting off-topic content before the given input text and identifies LLM-generated text based on an established detection threshold. Experimental results on both public and generated datasets demonstrate that Short-PHD outperforms existing zero-shot methods in short LLM-generated text detection. The implementation codes of this study are available online.[1]

## 1 Introduction

Table 1: Detection performance (AUC) of PHD and Short-PHD for identifying short LLM-generated texts (50 tokens). Results are averaged from Table 4.

| Method | OPT/OPT-iml/GPT-NeoX | GPT-3.5/GPT-4o |
|---|:---:|:---:|
| PHD (Tulchinskii et al., 2024) | 0.632 | 0.653 |
| Short-PHD (Ours) | **0.793** (Improve ↑ 25.47%) | **0.830** (Improve ↑ 27.11%) |

With the rapid development of Large Language Models (LLMs) (Guo et al., 2025), detecting LLM-generated content has become a crucial issue in controlling malicious content on the internet (Lin et al., 2022; Liu et al., 2024; Fang et al., 2024). In this work, we focus on zero-shot methods for detecting LLM-generated texts. A typical zero-shot method leverages an LLM as a scoring model to compute the conditional probability of each token given previous tokens and analyzes the statistical properties of these probabilities (Ippolito et al., 2020; Su et al., 2023; Mitchell et al., 2023). For example, DetectGPT (Mitchell et al., 2023) and Fast-DetectGPT (Bao et al., 2024) focus on conditional probability curvature, which evaluates the direction of perplexity change after perturbing random tokens. However, to ensure reliable performance, these methods often require a 'white-box' setting in which the generative LLM can serve as the scoring model during detection. For example, DetectGPT achieves a detection AUC of only 0.6 when using GPT-J as the scoring model to detect GPT-2-generated text (Mitchell et al., 2023).

To overcome the 'white-box' challenge, apart from probability-based methods, Tulchinskii et al. (2024) propose utilizing the Persistent Homology Dimension (PHD) from topological

---

*Dongjun Wei and Minjia Mao contribute equally. Corresponding to Minjia Mao.

[1] https://github.com/djwei96/ShortPHD

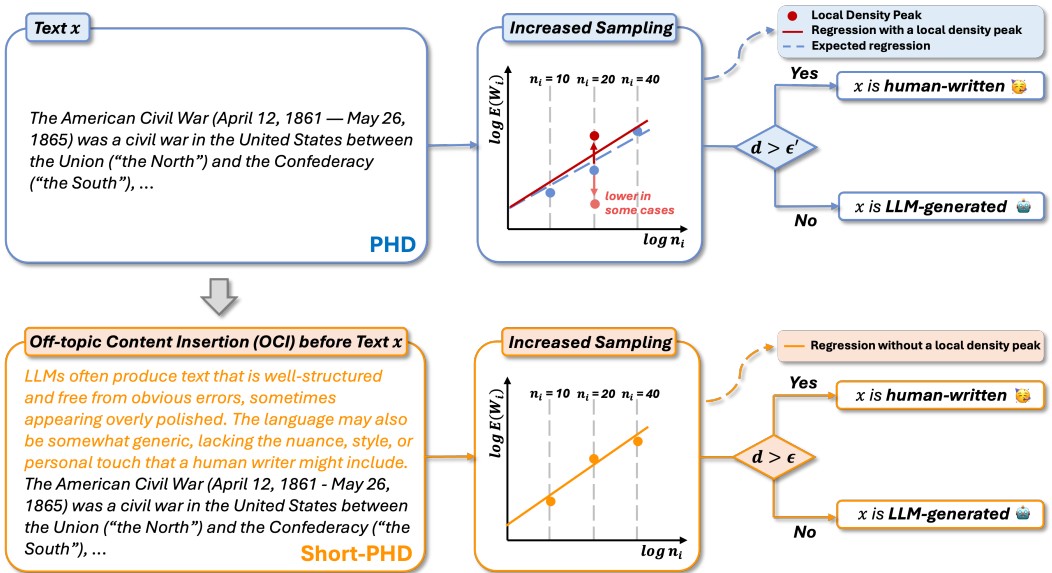

Figure 1: PHD (Tulchinskii et al., 2024) v.s. Short-PHD. $n_i$ is the number of sampled points in the point cloud $W$. $W_i$ is the sampled subset of points in $W$ with size $n_i$. $E(W_i)$ is computed by the minimal spanning tree (MST).

data analysis to identify LLM-generated texts. Specifically, texts are first embedded using a pretrained LLM such as RoBERTa (Liu et al., 2019). Next, the PHD (Kališnik, 2019), a concept of topological data analysis (Chazal & Michel, 2021), is computed for these embeddings. From a topological perspective, PHD measures the connectedness of the embedding, and a higher PHD represents a less connected data structure. **LLM-generated texts are found to exhibit lower PHD values than human-written text** due to their naturally higher connectedness in the embedding space. PHD outperforms other zero-shot methods, such as DetectGPT, in detecting LLM-generated texts across various models and domains, and also demonstrates greater robustness against adversarial attacks (Tulchinskii et al., 2024).

However, similar to other zero-shot methods, the PHD method can only detect text that is sufficiently long, and Tulchinskii et al. (2024) assume LLMs are fluent enough to generate at least 200 tokens, which limits their applicability for detecting short texts. In this study, we first analyze the scalability challenges of the current PHD method. Specifically, based on the increased sampling technique used to estimate PHDs, the PHD method encounters a phenomenon named local density peaks, which can make estimation unstable (either too high or too low). These local density peaks are particularly problematic for short texts, as they lack sufficient variability to sample out of the peak.

To facilitate the PHD method to overcome the local density peak issue in short texts, we propose a hypothesis involving an off-topic content insertion (OCI) technique. Specifically, we hypothesize that **inserting off-topic content unrelated to the input text can stabilize and increase the PHD score**. The underlying rationale is that by adding extra content, the text becomes longer, helping the PHD estimation avoid local density peaks in increased sampling. Furthermore, since PHD measures the connectedness of embeddings, limiting the inserted content to off-topic (i.e., irrelevant) content ensures that the PHD increases for all inputs because text embeddings are less connected.

Based on OCI, we present Short-PHD, a zero-shot method derived from the original PHD method, specifically tailored to detect short LLM-generated texts. A comparison between Short-PHD and PHD is illustrated in Figure 1. Specifically, given a short human-written text and a short LLM-generated text, Short-PHD first inserts the same off-topic content into each text. Then, it applies increased sampling to compute their respective PHD scores. According to the hypothesis on OCI, introducing OCI stabilizes and increases the PHD values of both

human-written and LLM-generated texts. To preserve the effectiveness of detection, we further hypothesize that **the gap between human-written and LLM-generated PHD values remains the same in expectation after inserting the same off-topic content**. Consequently, Short-PHD can establish a threshold of PHD to identify LLM-generated texts.

We summarize our main contributions as follows: 1) We identify the local density problem in estimating PHDs of short texts in the previous PHD method and propose that OCI could alleviate this issue. 2) We propose a novel Short-PHD method based on OCI to detect short LLM-generated texts. 3) Experimental results on both public and generated datasets indicate that Short-PHD achieves state-of-the-art performance in detecting short LLM-generated texts compared to the previous PHD method and other baselines.

## 2 Preliminary

The persistent homology dimension (PHD) analyzes the data in high dimensional space about the birth and death of its persistence cycles (Tulchinskii et al., 2024). Empirical evidence shows that LLM-generated texts exhibit lower PHD values than human-written texts. Formally, let $\mathcal{W} \subset \mathbb{R}^d$ be a $d$-dimensional topological space, and let $W = \{w_i \in \mathbb{R}^d\}$ be a discrete point cloud sampled from $\mathcal{W}$, i.e., $W \subset \mathcal{W} \subset \mathbb{R}^d$. In our work, we represent text as a point cloud in a high-dimensional space using embeddings from a pretrained LLM such as LLaMA (Dubey et al., 2024). Specifically, given a text sequence of $n$ tokens, each token is mapped to a $d$-dimensional embedding vector, forming a set of $n$ points in $\mathbb{R}^d$. For example, with $d = 768$, the text forms a point cloud in $\mathbb{R}^{768}$, where each point corresponds to a token embedding.

### 2.1 Mathematics Preliminary

We provide basic concepts of homology groups in Appendix A.1. Informally speaking, the idea of homology groups is to find non-trivial topological features in a topological space. These features include connected components, holes, and cavities that exist in the space. Next, we introduce the mathematical concepts of persistent homology in Appendix A.2. To sum up, different from homology groups that analyze a topological space, persistent homology belongs to data analysis that analyzes data points (or a point cloud). First, from the point cloud $W$, the Vietoris-Rips (VR) complexes VR($W$) (Bakhtin et al., 2019) are constructed with a filtrating parameter $r$. By changing the filtrating parameter $r$, persistent homology focuses on the birth and death of cycles ($t_{\text{birth}}, t_{\text{death}}$) in the complex. **The persistent homology dimension (PHD) is defined based on the lifetime of these cycles.** We refer readers to Appendix A.2 and previous work (Birdal et al., 2021) for more details.

### 2.2 Estimating Persistence Homology Dimension

Following previous work (Tulchinskii et al., 2024), we focus on $\text{PH}_0$, which measures the 0-persistent homology. The $\text{PH}_0$ corresponds to the minimal spanning tree (Tulchinskii et al., 2024) and Equation A1 (in Appendix A.2) can be calculated as

$$E_\alpha^0(W) = \sum_{e \in \text{MST}(W)} |e|^\alpha, \tag{1}$$

where MST($W$) denotes the minimal spanning tree (MST) from point set $W$, $e$ is the edge in the tree, and $|e|$ is the length of $e$. The exact number of $\text{PH}_0$ is hard to compute even for a small point cloud, but it could be approximated through increased sampling. The increased sampling is conducted by sampling different numbers of data points in the point cloud and estimating their MST. Specifically, if points $w_i$ in a cloud are independently distributed, then with probability approaching one, $E_\alpha^0(W) \sim Cn^{\frac{d-\alpha}{d}}$ holds as $n \to \infty$ (Steele, 1988). Therefore, in practice, by increasing the number $n_i$ of sampled points in a subset $W_i = \{w_1, w_2, \ldots, w_{n_i}\}$, we can find its minimal spanning tree and compute $E_\alpha^0(W_i)$. Since we know that $\log E_\alpha^0(W_i) \sim \left(1 - \frac{\alpha}{d}\right) \log n_i + \log C$, we can fit a linear regression to compute

$d$ if we fix $\alpha$ (usually $\alpha \in [0.5, 2.5]$). We write the PHD calculation as $d = \text{PHD}(W)$ to simplify our notation.

**Problems in Estimating PHD for Short Texts (Local Density Peak).** Although estimating PHD with increased sampling is sample-efficient, it introduces additional challenges for short texts, as their point cloud contains only a small number of points. When the number of points is small, there is a higher probability that these points are too close (or distinct) to each other. This could cause $\log E_\alpha^0(W_i)$ to become unstable due to the edge lengths of the MST becoming extremely small (or large). We refer to this issue as a local density peak (Tulchinskii et al., 2024). Meanwhile, as previously discussed, the value of PHD is based on the slope of the linear regression between $\log n_i$ and $\log E_\alpha^0(W_i)$. Therefore, an unstable $\log E_\alpha^0(W_i)$ caused by local density peaks could affect the accuracy of the PHD estimation process. An example is shown in Figure 1. Specifically, the regression contains three sampled subsets through increased sampling, which contain 10, 20, and 40 points (before logarithm), where $\log 20$ experiences a local density peak. The peak will lead to a higher slope and thereby lead to a higher PHD value $d$. In practice, the increased sampling process is conducted multiple times to reduce the influence of local density peaks (Birdal et al., 2021). However, if we only have a few points to sample, the local density peak cannot be easily avoided by the randomness in sampling.

## 3   Method: Short-PHD

In this section, we first introduce a technique named off-topic insertion (OCI) which can stabilize the PHD estimation for short texts. We then present Short-PHD, an OCI-based method for detecting short LLM-generated texts. Finally, we provide empirical evidence to validate the effectiveness of these two techniques.

### 3.1   OCI: Off-topic Content Insertion

As discussed in Section 2.2, compared to long texts, the PHD method is more likely to encounter local density peaks when estimating short texts, leading to unstable results that may be either too high or too low. To address such an issue, we propose a hypothesis using off-topic insertion as follows:

**Hypothesis 1 (OCI).** *Off-topic content insertion (OCI) will stabilize and increase the PHD values of short texts.*

The rationale behind this hypothesis is twofold. First, by adding additional content, the text becomes longer, helping the PHD estimation avoid local density peaks and achieve more stable results. Second, since PHD measures the connectedness of the embedding, adding completely off-topic text reduces this connectedness, thereby increasing the PHD for any given input text. For example, if the original text is a Wikipedia page on the American Civil War, the off-topic content could be a tip for detecting LLM-generated texts. In practice, we insert a fixed text sequence before each text sample. We apply OCI in distinguishing short human-written and LLM-generated texts below.

### 3.2   Short-PHD: Detecting Short LLM-generated Text

We assume that the PHD distributions of short human-written and LLM-generated texts follow independent normal distributions, i.e., $d^h \sim N(\mu^h, (\sigma^h)^2)$ and $d^m \sim N(\mu^m, (\sigma^m)^2)$, where the superscripts $h$ and $m$ signify human and machine, respectively. Following Tulchinskii et al. (2024), human-written texts exhibit larger PHDs than LLM-generated texts. Therefore, we can use PHD to detect LLM-generated text by measuring the probability that a human-written text has a higher PHD than an LLM-generated text, i.e., $P(d^h > d^m)$. Specifically, let $Z = d^h - d^m$. Then, the probability of $d^h$ being greater than $d^m$ is given by $P(d^h > d^m) = P(Z > 0)$. Based on our assumption, $d^h$ and $d^m$ are independent, then $Z$ follows a normal distribution $Z \sim N(\mu^h - \mu^m, (\sigma^h)^2 + (\sigma^m)^2)$. Thus, the standardized $Z$ follows a standard normal distribution. The probability of correctly classifying a text is

---

**Algorithm 1** Short-PHD

---

1: **Input:** a text sequence $X$, an off-topic content set $T_X$, an pretrained LLM $P_M$, a PHD estimator PHD(), and a detection threshold $\epsilon$.
2: Initialize a score list $S$
3: **for** each off-topic content $x_t \in T_X$ **do**
4:      $X_t \leftarrow \text{concat}(x_t, X)$
5:      Compute the embedding of $X_t$, $W_t \leftarrow P_M(X_t)$
6:      Estimate the PHD of $W_t$, $d_t \leftarrow \text{PHD}(W_t)$
7:      Record $d_t$ in $S$
8: **end for**
9: Score $s \leftarrow \text{mean}(S)$
10: **if** $s > \epsilon$ **then**
11:      return human-written
12: **else**
13:      return LLM-generated
14: **end if**

---

given by

$$P(Z > 0) = \Phi\left( (\mu^h - \mu^m) / \sqrt{(\sigma^h)^2 + (\sigma^m)^2} \right), \tag{2}$$

where $\Phi$ is the cumulative distribution function (CDF) of a standard normal distribution. Since $\Phi$ is monotonically increasing, a larger mean difference $(\mu^h - \mu^m)$ and lower variance $((\sigma^h)^2 + (\sigma^m)^2)$ increase the probability of successfully detecting LLM-generated text.

When the text is short, the estimation of PHD becomes unstable, and the variance $((\sigma^h)^2 + (\sigma^m)^2)$ increases, making detection ineffective. Therefore, in Short-PHD, we apply OCI to stabilize the estimation process for detection. Similarly to the previous PHD method analysis, we assume that each PHD distribution after OCI follows independent normal distributions: $d_I^h \sim N(\mu_I^h, (\sigma_I^h)^2)$ and $d_I^m \sim N(\mu_I^m, (\sigma_I^m)^2)$, where $I$ denotes OCI. According to Hypothesis 1 on OCI, the variance of PHD decreases $(\sigma_I^h < \sigma^h$ and $\sigma_I^m < \sigma^m)$, while the PHD values for both human-written and LLM-generated texts increase. We approximate this increase as follows $\mu_I^h \approx \mu^h + \epsilon^h$ and $\mu_I^m \approx \mu^m + \epsilon^m$, where $\epsilon^h$ and $\epsilon^m$ represent the increases in PHD for human-written and LLM-generated texts, respectively. In Short-PHD, we further make the following assumption on the increase.

**Hypothesis 2 (Short-PHD).** *The gap between human-written and LLM-generated PHD values remains the same in expectation after inserting the same off-topic content.*

This implies that $\mu^h - \mu^m \approx \mu_I^h - \mu_I^m$, which means the PHD scores of human-written and LLM-generated texts increase by the same value $\epsilon^h \approx \epsilon^m$. Therefore, in Short-PHD, when the gap between human-written and LLM-generated texts remains and the variances are decreased, we have

$$P(d^h > d^m) < P(d_I^h > d_I^m), \tag{3}$$

meaning that LLM-generated texts can be detected with a higher probability using Short-PHD compared to PHD. We present Short-PHD in Figure 1 and Algorithm 1. In practice, Short-PHD employs OCI multiple times with different content and computes the average.

### 3.3 Empirical Validation

We conduct an empirical validation to test the proposed hypotheses on OCI and Short-PHD using the same dataset from the original PHD paper (Tulchinskii et al., 2024). Specifically, we choose two subsets from Tulchinskii et al. (2024): Reddit WritingPrompts (WP) stories (Fan et al., 2018) and Wikipedia pages (Guo et al., 2020), both of which contain GPT-3.5-generated and human-written texts. We leverage LLaMA-3-8B (Dubey et al., 2024) to compute the embedding $W = \text{LLaMA}(X)$ of a text sequence $X$ and calculate the PHD detection score using increased sampling. We then apply OCI before the input text and calculate the PHD

Table 2: Average number of tokens in each dataset and statistics of detection score distributions in each dataset with PHD and Short-PHD. Bolded standard deviation (std) implies large std reduction by Short-PHD.

| Dataset | WP human | WP GPT-3.5 | Wikipedia human | Wikipedia GPT-3.5 |
|---|---|---|---|---|
| Token number | 153 | 276 | 461 | 165 |
| PHD mean | 10.17 | 8.84 | 10.70 | 8.93 |
| PHD std | 1.49 | 0.79 | 1.01 | 1.54 |
| Short-PHD mean | 10.26 | 9.23 | 10.81 | 9.23 |
| Short-PHD std | **0.79** | 0.67 | 0.85 | **1.00** |

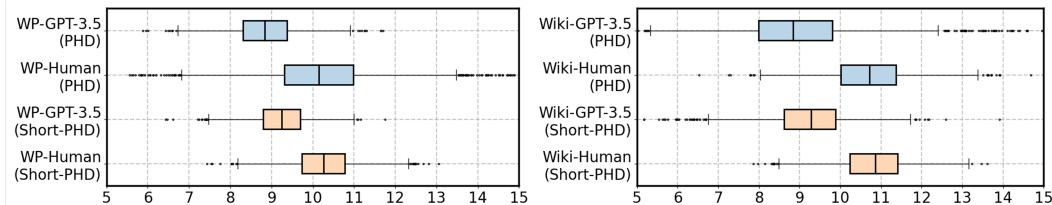

Figure 2: Boxplots of score distributions on human-written and GPT-3.5-generated Wikipedia and WritingPrompts (WP) data.

score for the modified text. Specifically, we choose off-topic content related to tips for detecting LLM-generated texts, which is generally unrelated to the text being detected. The off-topic content is shown in Figure 1. We provide the average number of tokens in each subset and the statistics of PHD scores in Table 2. Additionally, we present the distributions of PHD scores using boxplots in Figure 2.

**Validation of Hypothesis 1 (OCI).** From Table 2, we first observe that the average token number of human-written WP and GPT-3.5-generated Wikipedia is significantly smaller than that of human-written Wikipedia and GPT-3.5-generated WP. Meanwhile, we also find that subsets with shorter texts have a larger standard deviation than those with longer texts. Similarly, as depicted in Figure 2, it is demonstrated that human-written WP and GPT-3.5-generated Wikipedia have wider (blue) boxes compared to other blue boxes, indicating larger variances. Therefore, we can validate that the PHD estimations are less stable for both short human-written and short LLM-generated texts than for long texts.

Next, as shown in Table 2, compared to the standard deviation of PHD, applying OCI significantly reduces the standard deviation of PHD values for both short human-written WP and GPT-3.5-generated Wikipedia. The same results can be concluded from Figure 2, where the yellow boxes (Short-PHD) of human-written WP and GPT-3.5-generated Wikipedia are narrower than the blue boxes of these two subsets. In summary, we conclude that OCI stabilizes the estimation of PHD for short texts. Second, we observe that the mean PHD values for all subsets simultaneously increase. Therefore, we also conclude that applying OCI increases the PHD values for both short human-written and LLM-generated texts.

**Validation of Hypothesis 2 (Short-PHD).** As depicted in Figure 2, the gap between human-written and LLM-generated text after applying OCI is largely maintained compared to the PHD method, without dropping significantly. For example, in the WP dataset, the gap for PHD is 1.33 $(10.17 - 8.84)$, while the gap for Short-PHD is 1.03 $(10.26 - 9.23)$. Similarly, in the Wikipedia dataset, the gap for PHD is 1.77 $(10.70 - 8.93)$, whereas for Short-PHD, it is 1.58 $(10.81 - 9.23)$. Therefore, the Short-PHD method could establish a detection threshold like the PHD method to identify LLM-generated texts. Furthermore, by plugging in the mean and standard deviation values, we can explicitly compute the value of the CDF. For the PHD method, the probabilities of successful detection are approximately $\Phi(\frac{10.17-8.84}{\sqrt{1.49^2+0.79^2}}) \approx \Phi(0.79) \approx 0.78$ and $\Phi(0.96) \approx 0.83$ for the WP and Wikipedia datasets, respectively. After applying Short-PHD, the probabilities increase to $\Phi(0.98) \approx 0.84$ and

$\Phi(1.20) \approx 0.88$ for the WP and Wikipedia datasets, respectively. This indicates that the detection performance of Short-PHD is improved. However, note that a PHD distribution does not strictly follow a normal distribution. These statistical results should only be considered an approximation.

## 4 Experiments

In this section, we evaluate our Short-PHD method using both public and generated datasets. We benchmark our method against various zero-shot and supervised detection approaches. Additionally, we conduct multiple sensitivity analyses for Short-PHD and evaluate its robustness against several detection attacks.

### 4.1 Experimental Setup

**Public Dataset Setting.** We first evaluate the Short-PHD method on the original dataset used in the PHD method (Tulchinskii et al., 2024), referred to as GPTID. The dataset consists of two domains of human-written text: Wiki40b (Guo et al., 2020) from Wikipedia and WritingPrompts from Reddit's stories subsector (Fan et al., 2018). The paper claims another StackExchange subset, but it is not publicly available. Text generation is performed using GPT-2 XL (Radford et al., 2019), OPT-13B (Zhang et al., 2022), and GPT-3.5. Detailed statistics are summarized in Table A1.

**Generated Dataset Setting.** We then construct a more comprehensive dataset in two additional domains using more diverse and advanced LLMs to further evaluate the performance of our method. Specifically, we select Wikihow (Koupaee & Wang, 2018) for articles and quizzes and Eli5 (Fan et al., 2019) for question answering. We employ OPT-13B (Zhang et al., 2022), OPT-iml-1.3B (Iyer et al., 2022), GPT-NeoX (Black et al., 2022), GPT-3.5-turbo, and GPT-4o for text generation. For each domain, we randomly sample 500 human-written texts and prompt each LLM to generate text. Detailed generation settings are provided in Appendix B.1.

**Baselines.** For zero-shot methods, we mainly compare our Short-PHD method with other state-of-the-art approaches, including PHD (Tulchinskii et al., 2024), DetectGPT (Mitchell et al., 2023), and DetectLLM (Su et al., 2023). Additionally, we evaluate Short-PHD against methods using heuristic scores, including Perplexity (mean log probabilities) and LogRank (average log of ranks). For supervised methods, we incorporate a fine-tuned RoBERTa model trained on GPT-2 outputs from OpenAI (Solaiman et al., 2019). We also implemented more recent baselines, including Fast-DetectGPT (Bao et al., 2024) and DNA-GPT (Yang et al., 2024) in Appendix B.

**Implementation Details.** We truncate human-written and LLM-generated texts to 50, 100, and 150 tokens to evaluate detection performance for short texts. For the Short-PHD method, we perform OCI 12 times using different off-topic content and utilize LLaMA-3-8B (Dubey et al., 2024) to compute the embedding of each insertion (i.e., the 5th line in Algorithm 1). For a fair comparison, we also leverage LLaMA-3-8b as the scoring model for all zero-shot baselines. The RoBERTa model is directly taken from OpenAI's public version without further modification. More implementation details are provided in Appendix B.1.

### 4.2 Main Results

We first conduct an empirical evaluation of our method compared to PHD using the public GPTID dataset employed by PHD (without truncation) in terms of AUC and TPR@5%FPR, as shown in Table 3. We observe that Short-PHD outperforms PHD across all four subsets in terms of both metrics. Next, we demonstrate the AUC scores on the generated dataset in Table 4 for Short-PHD and baselines. As reported, our Short-PHD method outperforms other baselines in almost all datasets across five LLMs and two domains, corroborating its effectiveness. Specifically, we observe that detecting short LLM-generated text is generally a challenging task for existing zero-shot methods. For example, DetectGPT fails to detect LLM-generated ELI5 answers, achieving an average AUC of only 0.24. This suggests

Table 3: Results on the GPTID dataset. The table reports the AUC and TPR@5%FPR (T@F) of PHD and Short-PHD. For results of other baselines, see Tulchinskii et al. (2024).

| | Wikipedia | | | | | | WritingPrompts | |
| | GPT-2 | | OPT | | GPT-3.5 | | GPT-3.5 | |
| Method | AUC | T@F | AUC | T@F | AUC | T@F | AUC | T@F |
|---|---|---|---|---|---|---|---|---|
| PHD | 0.825 | 0.388 | 0.821 | 0.423 | 0.847 | 0.549 | 0.808 | 0.088 |
| Short-PHD | **0.860** | **0.445** | **0.854** | **0.446** | **0.891** | **0.555** | **0.845** | **0.329** |
| Improve(↑) | 4.242 | 14.691 | 4.019 | 5.437 | 5.195 | 1.093 | 4.579 | 273.864 |

Table 4: Results on the Generated Dataset. The table reports AUC of baselines and our method on texts (50 tokens). The improvement is computed between PHD and Short-PHD.

| | Wikihow | | | | | Eli5 | | | | |
| Method | OPT | OPT-iml | NeoX | GPT-3.5 | GPT-4o | OPT | OPT-iml | NeoX | GPT-3.5 | GPT-4o |
|---|---|---|---|---|---|---|---|---|---|---|
| RoBERTa | 0.66 | **0.77** | 0.66 | 0.65 | 0.66 | 0.75 | 0.78 | 0.77 | 0.80 | 0.80 |
| Perplexity | 0.47 | 0.44 | 0.47 | 0.62 | 0.63 | 0.46 | 0.37 | 0.46 | 0.70 | 0.73 |
| LogRank | 0.46 | 0.43 | 0.47 | 0.58 | 0.59 | 0.42 | 0.33 | 0.42 | 0.62 | 0.64 |
| DetectGPT | 0.56 | 0.53 | 0.56 | 0.62 | 0.59 | 0.22 | 0.16 | 0.21 | 0.27 | 0.27 |
| DetectLLM | 0.44 | 0.43 | 0.41 | 0.45 | 0.47 | 0.37 | 0.30 | 0.34 | 0.36 | 0.36 |
| PHD | 0.62 | 0.60 | 0.66 | 0.66 | 0.63 | 0.64 | 0.65 | 0.62 | 0.66 | 0.66 |
| Short-PHD | **0.78** | 0.73 | **0.77** | **0.80** | **0.79** | **0.84** | **0.82** | **0.82** | **0.87** | **0.86** |
| Improve(↑) | 25.81 | 21.67 | 16.67 | 21.21 | 25.40 | 31.25 | 26.15 | 32.26 | 31.82 | 30.30 |

that probability alone is not a reliable metric for detecting short texts. In contrast, PHD (Tulchinskii et al., 2024) attains an average AUC of approximately 0.65. Similarly, our method achieves an average AUC score of over 0.8, surpassing even the supervised RoBERTa detector. We thus conclude that PHD captures more meaningful information for short text detection through text embeddings. Furthermore, we present box plots of PHD distributions computed by PHD and Short-PHD in Figure 3 for texts generated by GPT-4o. The results for all LLMs are provided in Figure A1 in the appendix. From these figures, we observe that OCI used by Short-PHD increases PHD's value and stabilizes PHD estimations by reducing variance. Meanwhile, the gap between human-written and LLM-generated texts remains largely unchanged, making Short-PHD effective for detection. As a result, Table 4 shows that our method outperforms PHD by more than 16% across all subsets.

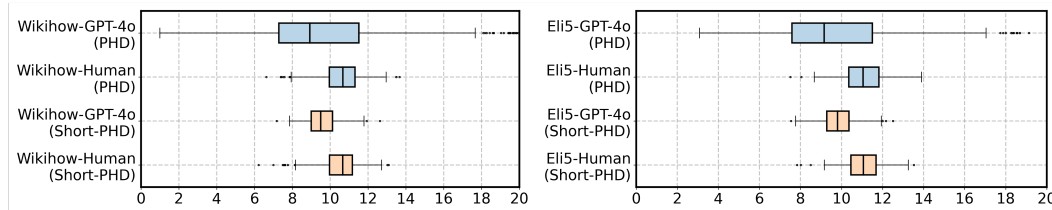

Figure 3: Box plots of PHD and Short-PHD distributions for texts generated by GPT-4o.

### 4.3 Sensitivity Analysis

We then conduct a series of sensitivity analyses regarding text length, OCI, white-box detection, and detection attacks to further evaluate Short-PHD.

**Sensitivity Analysis on Text Length.** We first evaluate the detection performance on texts with a fixed length of 50, 100, and 150 tokens in Figures 4a and 4b. As depicted, for texts with a fixed length of 50 tokens, Short-PHD significantly outperforms PHD in terms of AUC. When the text becomes longer, at 100 and 150 tokens, the gap continues to narrow,

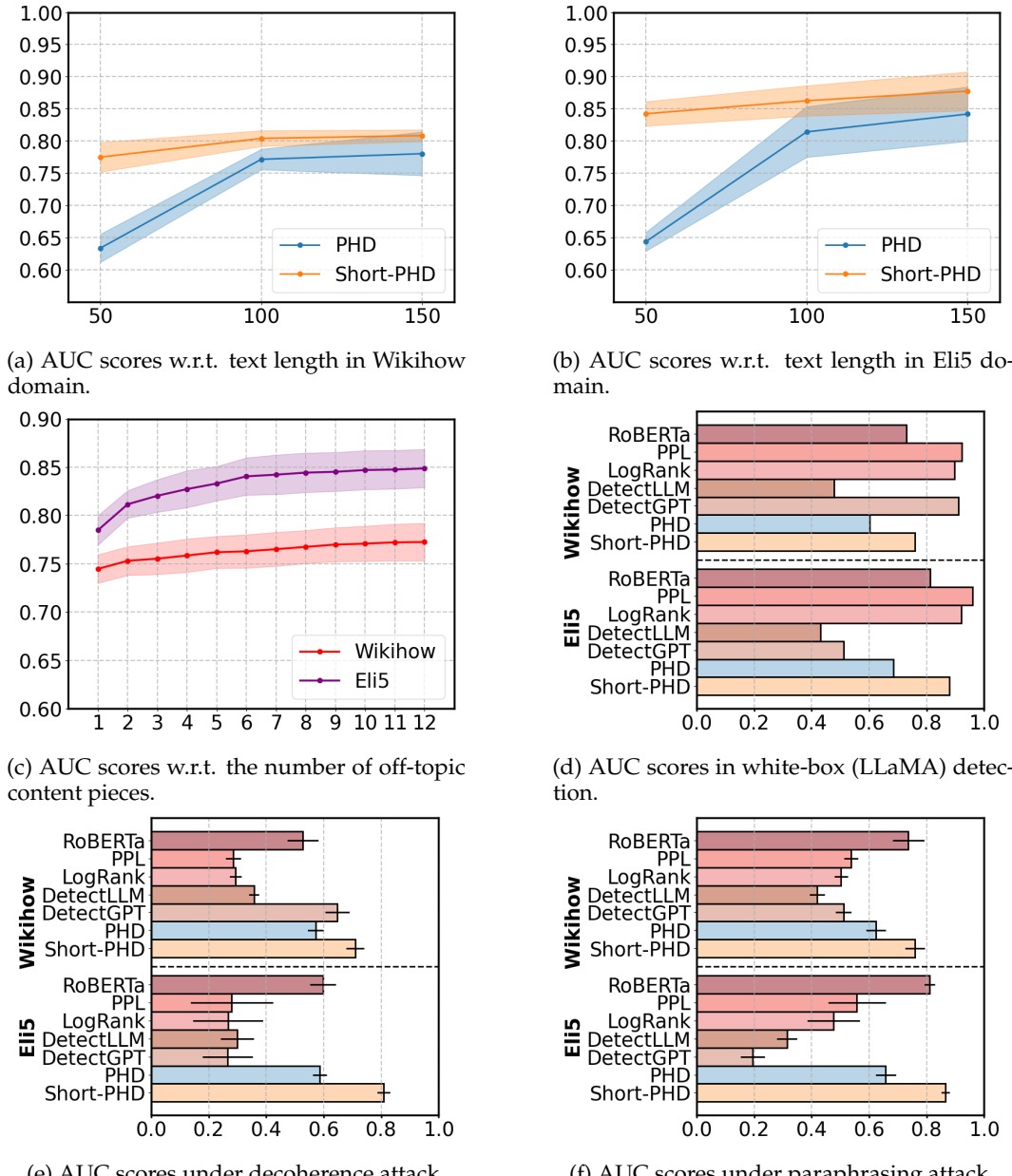

(a) AUC scores w.r.t. text length in Wikihow domain.

(b) AUC scores w.r.t. text length in Eli5 domain.

(c) AUC scores w.r.t. the number of off-topic content pieces.

(d) AUC scores in white-box (LLaMA) detection.

(e) AUC scores under decoherence attack.

(f) AUC scores under paraphrasing attack.

Figure 4: Sensitivity Analysis.

but Short-PHD still surpasses PHD. The reason is that as the text length increases, the PHD method encounters fewer local density peak problems. To sum up, our proposed Short-PHD is highly effective for short texts of around 50 tokens and remains superior to the original PHD method for longer texts.

**Sensitivity Analysis on the Number of Off-topic Content Pieces.** We adjust the number of content pieces in the off-topic content set in Algorithm 1 and analyze its impact. As depicted in Figure 4c, by simply inserting a piece of off-topic content, the average AUC score reaches around 0.75. The performance of Short-PHD stabilizes when at least 6 content pieces are provided. This demonstrates that by inserting multiple off-topic content pieces, the performance of Short-PHD becomes more stable and effective.

Table 5: Sensitivity Analysis on the Topics of the Inserted Content.

| | Wikihow | | | | | Eli5 | | | | |
|---|---|---|---|---|---|---|---|---|---|---|
| | OPT | OPT-iml | NeoX | GPT-3.5 | GPT-4o | OPT | OPT-iml | NeoX | GPT-3.5 | GPT-4o |
| On-topic | 0.66 | 0.67 | 0.73 | 0.60 | 0.60 | 0.60 | 0.60 | 0.60 | 0.56 | 0.55 |
| Off-topic | **0.78** | **0.73** | **0.77** | **0.80** | **0.79** | **0.84** | **0.82** | **0.82** | **0.87** | **0.86** |

**Sensitivity Analysis on White-box Detection.** We then evaluate the performance of each method in a white-box setting. Specifically, since we use LLaMA as the scoring model for all cases, we generate WikiHow and ELI5 articles using LLaMA with 50 tokens and then evaluate the detection performance. As shown in Figure 4d, probability-based methods such as PPL and DetectGPT achieve the best performance. This is because the probability calculation is more accurate in a white-box scenario. However, the Short-PHD method still achieves an AUC score above 0.9, outperforming PHD and baselines like RoBERTa and DetectLLM, demonstrating its effectiveness in the white-box setting.

**Sensitivity Analysis on the Topics of the Inserted Content.** We then evaluate how the relevance of the inserted topic affects the detection performance of the proposed method, considering both off-topic and on-topic content. For off-topic content, we follow the same setup as in the main experiment. For on-topic content, we apply a simple heuristic approach by inserting the same content as the detected text. This ensures that the inserted content is on the same topic as the detected text. The results shown in Table 5 demonstrate that inserting off-topic content yields higher detection scores across all generators and domains compared to inserting the same detected text. It provides some initial evidence that simply increasing the length of the detected text does not improve detection.

**Robustness against Detection Attacks.** Lastly, we evaluate the robustness of each method against the decoherence attack (Bao et al., 2024) and the paraphrasing attack (Krishna et al., 2023) (see Appendix B.2 for more details) in Figure 4e and Figure 4f. As shown in both figures, we find Short-PHD is robust against these attacks, which is consistent with previous work, which shows PHD is more robust compared to other baselines (Tulchinskii et al., 2024). Similar to Tulchinskii et al. (2024), we observe that the detection AUCs are almost the same before and after attacks, demonstrating the robustness of Short-PHD.

## 5 Conclusions and Future Work

In this study, we propose Short-PHD, a novel zero-shot LLM-generated text detection method tailored for short texts. Experimental results demonstrate that Short-PHD achieves superior detection performance for short texts compared to PHD and other baselines. Meanwhile, Short-PHD is robust against various detection attacks.

We acknowledge several limitations of our study. First, the original PHD method is employed in different languages, but we only test Short-PHD in English. Second, we utilize only persistent homology in 0 dimensions. Whether high-dimensional topological structures of embeddings can be used for detection remains to be explored. Third, note that PHD can be viewed as a feature of the text. Future work can take PHD into account when developing training-based methods.

## 6 Related Work

Existing methods for detecting LLM-generated texts can be divided into two categories: supervised detectors and zero-shot methods (Wu et al., 2023). In this work, we focus on zero-shot detection methods, which require no training and exhibit better generalizability in identifying LLM-generated text across various models and domains (Bao et al., 2024). We refer readers to Appendix C for a detailed review of each stream of literature.

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

# Appendix A  Mathematics Preliminary

## A.1  Homology Group

Informally speaking, the idea of homology groups is to find non-trivial topological features in a topological space. These features include connected components, holes, and cavities that exist in the space.

Concretely, if $\mathcal{W}$ is a simplicial complex, for $k = 1, 2, \ldots$, a proper boundary operator (group homomorphism) can be defined as $\partial_k : C_k(\mathcal{W}) \to C_{k-1}(\mathcal{W}), \forall k$, where $C_k(\mathcal{W})$ is the free abelian group generated by oriented $k$-simplices in $\mathcal{W}$. Next, we can form a chain of group homomorphism: $\cdots \to C_{k+1}(\mathcal{W}) \xrightarrow{\partial_{k+1}} C_k(\mathcal{W}) \xrightarrow{\partial_k} C_{k-1}(\mathcal{W}) \to \cdots \to C_0(\mathcal{W})$, with $\partial_k \circ \partial_{k+1} \equiv 0$. In this work, we consider finite $\mathcal{W}$ because the number of tokens LLMs can embed has a maximum limit in practice (e.g., 768 for LLaMA).

Concretely, the $k$-th homology group $H_k(\mathcal{W})$ is defined as the quotient group $H_k(\mathcal{W}) = Z_k(\mathcal{W})/B_k(\mathcal{W})$, where $Z_k(\mathcal{W}) = \ker(\partial_k)$ is the cycle group and $B_k(\mathcal{W}) = \text{im}(\partial_{k+1})$ is the boundary group. The rank of the $k$-th homology group is a topological invariance, known as the Betti number $\beta_k = \text{rank}(H_k)$. Informally and geometrically speaking, the homology group characterizes the 'closed but non-boundary' structure.

## A.2  Persistent Homology Dimension

**Persistent Homology.** The idea of persistent homology is to construct complexes from a point cloud. Denote $B_\delta(x)$ as the ball with center $x$ and radius $\delta$ in $R^d$. A widely-used Vietoris-Rips (VR) complex is defined as follows (Bakhtin et al., 2019).

**Definition 1 (VR complex).** Given a set of points $W$ and a radius $r$, a subset $Q \subset W$ is constructed as an element of the VR complex using the intersection of pairwise balls: $W_r = \text{VR}_r(W) = \{Q \subset W : \forall x, x' \in Q, B_r(x) \cap B_r(x') \neq 0\}$.

The persistent homology is defined based on filtrating certain complexes constructed from a set of points. Note that the filtrating parameter of VR complexes is the radius $r$. Each persistent homology $\text{PH}_k, k = 0, 1, 2, \ldots$ is a detailed analysis of homology in dimension $k$. Persistent homology focuses on the birth and death of cycles along with the filtrating parameter. Concretely, the $k$-persistent homology $\text{PH}_k$ keeps track of the lifespan of each $k$-persistence cycle $\gamma_i$. The lifespan is a pair of birth and death thresholds for each cycle $(t_{\text{birth}}, t_{\text{death}})$. For $k$-persistence cycle $\gamma_i$, denote $(t_{\text{birth}}(\gamma_i), t_{\text{death}}(\gamma_i))$ as its own lifespan features. Specifically, for VR complexes, $\text{PH}_k$ records the lifespan of $k$-persistence cycle as the radius increases.

For example, geometrically speaking, a 0-persistence cycle tracks the evolution of connected components in the topological space. The birth of each cycle is always 0 because each point starts as its own connected component unless there are identically the same points. By increasing the filtration parameter (radius), the death of the cycle happens when two connected components merge. The $\text{PH}_0$ is related to many geometrical meanings such as the maximal number of disjoint balls (Simsekli et al., 2020; Birdal et al., 2021) and minimal spanning tree (Bauer, 2021; Tulchinskii et al., 2024), which makes it easy to calculate.

**Persistent Homology Dimension.** There are many approaches to extract features from the birth and death thresholds (Pun et al., 2022). For example, previous work measures statistics of birth and death thresholds as features, including the first/second/third largest Betti numbers (Cang et al., 2015), min-plus and max-plus coordinates (Adcock et al., 2016; Kališnik, 2019). In this work, we follow previous work (Birdal et al., 2021) and define the $\alpha$-weighted $k$-th homology lifetime sum as follows

$$E_\alpha^k(W) = \sum_{\gamma_i \in \text{PH}_k(\text{VR}(W))} |I(\gamma_i)|^\alpha, \tag{A1}$$

where $I(\gamma_i) = t_{\text{death}}(\gamma_i) - t_{\text{birth}}(\gamma_i)$ is the lifetime of a cycle $\gamma_i$ in the $k$-th persistent homology of the VR complex $\text{PH}_k(\text{VR}(W))$. Next, the $k$-PHD is defined as

$$\dim_{\text{PH}}^k = \inf\{d|E_d^k(W') < C; \exists C > 0, \forall \text{ finite } W' \subset W\}, \tag{A2}$$

where $W'$ is a finite subset of $W$. Since $W$ is finite in this work, $W'$ can be any subset of $W$. By estimating the lifetime of cycles, PHD is a suitable estimator that combines both local and global features of the dataset.

## Appendix B  Experiments

Table A1: Statistics of the GPTID dataset.

| Domain | LLM | Number of pairs | Average human tokens | Average LLM tokens |
|---|---|---|---|---|
| Wikipedia | GPT-2 | 2557 | 465 | 272 |
| Wikipedia | OPT | 2475 | 463 | 260 |
| Wikipedia | GPT-3.5 | 2317 | 461 | 165 |
| WritingPrompts | GPT-3.5 | 2128 | 153 | 276 |

Table A2: The off-topic content set used in Algorithm 1.

| No. | Off-topic Content |
|---|---|
| 1 | Identifying text generated by large language models (LLMs) like GPT can be challenging, but there are some strategies and telltale signs that can help. Here are some suggestions: Repetition and Redundancy and Overly Polished or Generic Language. |
| 2 | LLM-generated text may contain repetitive phrases or ideas. Although models are improving, they might still circle back to the same points or restate them in slightly different ways. |
| 3 | LLMs often produce text that is well-structured and free from obvious errors, sometimes appearing overly polished. The language may also be somewhat generic, lacking the nuance, style, or personal touch that a human writer might include. |
| 4 | The tone or style of LLM-generated text might fluctuate unexpectedly. For instance, a paragraph may shift from formal to informal language or from being highly detailed to overly vague. |
| 5 | While LLMs are generally accurate, they can make mistakes that are unusual for humans, such as misinterpreting context, creating fictional facts, or making nonsensical statements. |
| 6 | LLMs can generate text that seems insightful but lacks true depth. The text may superficially address a topic without providing meaningful analysis or understanding. |
| 7 | The phrasing in LLM-generated text might sound slightly off, either too formal or too casual, or use uncommon word combinations that a human writer might not choose. |
| 8 | LLMs might draw connections between ideas that don't logically follow from each other. This can result in text that feels disjointed or where the flow of argument is not clear. |
| 9 | Some LLMs may overuse certain phrases or sentence structures, especially ones that were prominent in their training data. |
| 10 | Use AI detection tools designed to identify LLM-generated content. These tools analyze patterns in the text to predict whether it was written by a human or a machine. |
| 11 | LLMs may provide inconsistent answers when asked the same question in different ways. A human writer is more likely to maintain consistent opinions or facts across different contexts. |
| 12 | Using a combination of these strategies can help you more reliably identify LLM-generated text, although it's worth noting that as AI models improve, the distinction between human and machine-generated text is becoming increasingly subtle. |

### B.1  Experimental Details

**Datasets.** The summary statistics for the GPTID dataset are presented in Table A1. Specifically, we select Wikihow (Koupaee & Wang, 2018) for articles and quizzes and Eli5 (Fan et al., 2019) for question answering. We employ OPT-13B (Zhang et al., 2022), OPT-iml-1.3B (Iyer et al., 2022), GPT-NeoX (Black et al., 2022), GPT-3.5-turbo, and GPT-4o for text generation.

Table A3: Examples of Prompts Used in the Generated Dataset.

| Domain | Prompt |
|---|---|
| Wikihow | Please, generate a short wikihow article in 100 words or less from title 'How to Get Rid of Adchoices' and headline 'Download and install AdwCleaner., Download and install Malwarebyte Antimalware., Download and install Spybot Free Edition., Reboot your computer in Safe Mode., Run AdwCleaner., Ensure that all the results are checked., ..., Reboot your Mac.' |
| Wikihow | Please, generate a short wikihow article in 100 words or less from title 'How to Make No Sew Curtains2' and headline 'Measure your windows to decide how large your want each curtain panel to be., Choose your fabric, then wash, dry, and iron it, if necessary, to get rid of any shrinking and wrinkles., ..., Hang your curtains.' |
| Wikihow | Please, generate a short wikihow article in 100 words or less from title 'How to Account for Cost of Goods Sold' and headline 'Determine the beginning inventory value.This should always be the ending inventory value from the previous reporting period., Add the value of all inventory purchases., ..., Record the journal entry if you are using a perpetual inventory method.' |
| Eli5 | I will ask you a question. For this question, provide me a short answer in 100 words or less in a formal academic and scientific writing voice. Question: How Precisely Are Satellites put into orbit? Is it to the meter? |
| Eli5 | I will ask you a question. For this question, provide me a short answer in 100 words or less in an expert confident voice. Question: Why is the ocean sometimes really dark, like in the north Atlantic yet in some places a turquoise colour like in the Caribbean or South Pacific? |
| Eli5 | I will ask you a question. For this question, provide me a short answer in 100 words or less child's voice. The child has a limited vocabulary and limited knowledge so they may make grammatical as well as factual mistakes. Question: Why did the Soviets accept the Finnish peace agreement instead of installing the puppet government they had prepared, despite having (at great cost) crushed the Finnish army? |

For each domain, we randomly sample 500 human-written texts and prompt each LLM to generate text. The examples of prompts used in Wikiohw and Eli5 to generate the dataset are shown in Table A3. We maintain the default configuration, such as temperature, in each LLM during generation.

**Implementation Details.** All experiments were conducted on a single NVIDIA H100 GPU. The off-topic content set used in Algorithm 1 is listed in Table A2. Following (Tulchinskii et al., 2024), we set the $\alpha$ to 1.0 to calculate the PHD dimension. For supervised methods, we use the public version of the OpenAI detector from Hugging Face.[2] For zero-shot methods, we use the public implementations of Perplexity, LogRank, DetectGPT, and DetectLLM.[3] For dataset generation, we apply Transformers[4] and take the weights of OPT-13B, OPT-1.3B, and GPT-NeoX from Hugging Face.[5] For GPT-3.5-turbo and GPT-4, we utilize the API from OpenAI.[6]

**Attacks.** The decoherence attack randomly flips two adjacent tokens when a sentence exceeds 20 words (Bao et al., 2024). The paraphrasing attack employs DIPPER (Krishna et al., 2023) to rewrite each text. We set the lexical and order parameters as $L = 60$ and $O = 60$.

## B.2 Additional Experimental Results

**Main Results.** We present the box plots of PHD distributions for text generated by each LLM in each domain in Figure A1. Specifically, we show the PHD distributions computed by both PHD and Short-PHD.

Additionally, we implemented two more recent baselines, which are Fast-DetectGPT (Bao et al., 2024) and DNA-GPT (Yang et al., 2024), in our comparison. We conducted experiments on black-box detection in the tables below for texts with 50 tokens each. The setting is kept the same as in the previous experiment in Table 4. As shown in Table A5, Short-PHD consistently outperforms or matches the performance of the recent baselines. Specifically,

---

[2]https://huggingface.co/openai-community/roberta-base-openai-detector
[3]https://github.com/baoguangsheng/fast-detect-gpt
[4]https://pypi.org/project/transformers
[5]https://huggingface.co
[6]https://platform.openai.com

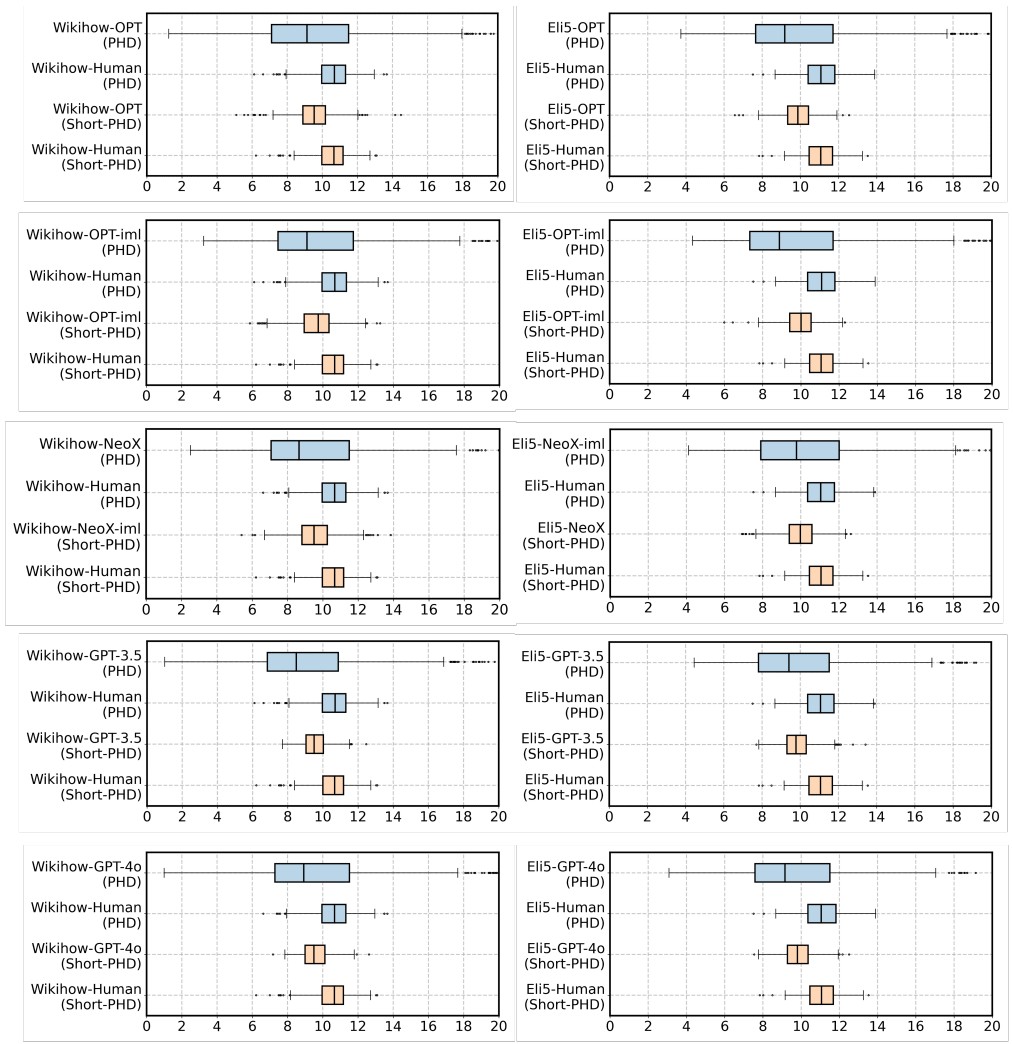

Figure A1: Box plots of PHD distributions by PHD and Short-PHD.

Table A4: Sensitivity Analysis.

(a) Sensitivity Analysis on the Number of Off-topic Content Pieces.

(b) Sensitivity Analysis on White-box Detection.

| Number | Wikihow | Eli5 |
|---|---|---|
| 1 | 0.745 | 0.785 |
| 2 | 0.753 | 0.812 |
| 4 | 0.759 | 0.827 |
| 6 | 0.763 | 0.840 |
| 8 | 0.768 | 0.844 |
| 10 | 0.771 | 0.847 |
| 12 (default) | 0.773 | 0.849 |

| Method | Wikihow | Eli5 |
|---|---|---|
| Roberta | 0.730 | 0.812 |
| Perplexity | 0.924 | 0.959 |
| LogRank | 0.897 | 0.921 |
| DetectGPT | 0.911 | 0.512 |
| DetectLLM | 0.478 | 0.431 |
| PHD | 0.602 | 0.684 |
| Short-PHD | 0.760 | 0.880 |

our method surpasses DNA-GPT significantly across different models and domains. Short-PHD also achieves an overall improvement in performance compared to Fast-DetectGPT for short text detection. These findings further highlight the effectiveness of short-PHD.

**Sensitivity Analysis.** We present the detailed results when changing the number of pieces of off-topic content in Table A4a. Meanwhile, we show the detailed white-box detection performance (AUC) in Table A4b. We find that for white-box detection, a simple perplexity

Table A5: Results on the Generated Dataset. The table reports AUC of baselines and our method on texts (50 tokens). For space concerns, Fast-D refers to Fast-DetectGPT.

| Method | Wikihow | | | | | Eli5 | | | | |
|---|---|---|---|---|---|---|---|---|---|---|
| | OPT | OPT-iml | NeoX | GPT-3.5 | GPT-4o | OPT | OPT-iml | NeoX | GPT-3.5 | GPT-4o |
| Fast-D | 0.73 | **0.74** | 0.74 | 0.77 | 0.76 | 0.82 | 0.81 | 0.80 | 0.83 | 0.84 |
| DNA-GPT | 0.57 | 0.55 | 0.58 | 0.64 | 0.64 | 0.55 | 0.50 | 0.55 | 0.73 | 0.74 |
| Short-PHD | **0.78** | 0.73 | **0.77** | **0.80** | **0.79** | **0.84** | **0.82** | **0.82** | **0.87** | **0.86** |

or log rank metric can achieve a satisfactory result because the probability is computed precisely.

## Appendix C   Related Work

Existing methods for detecting LLM-generated text can be categorized into supervised and zero-shot methods (Wu et al., 2023). Early supervised methods were trained on extracted linguistic or semantic features (Bhatt & Rios, 2021; Sadasivan et al., 2023; Li et al., 2024). Current supervised methods focus on training classifiers to distinguish the neural representations of LLM-generated text from human-written text (Liu et al., 2019; Zellers et al., 2019). Research has shown that supervised methods can easily overfit to the extracted features or texts in the training dataset (Bakhtin et al., 2019), making them prone to overfitting to the training distribution of specific LLMs and domains. Therefore, we focus on zero-shot detection methods, which require no training and exhibit better generalizability in identifying LLM-generated text across various models and domains (Bao et al., 2024).

Zero-shot methods focus on extracting certain features from large language models and set a threshold to differentiate between LLM-generated and human-written text. For example, one early detection method established a threshold based on the average log probability of a text, which serves as a strong baseline for detecting LLM-generated text (Solaiman et al., 2019). Recent works, such as DetectGPT (Mitchell et al., 2023) and Fast-DetectGPT(Bao et al., 2024), focus on conditional probability curvature, which evaluates the direction of perplexity change after perturbing random tokens. In this work, we investigate a novel detection score called persistent homology dimension (PHD) (Birdal et al., 2021; Tulchinskii et al., 2024), which shows that LLM-generated text tends to have a lower PHD value than human-written text. However, prior studies on PHD face challenges with short texts due to local density peak issues. We address this challenge by proposing a novel method called off-topic content insertion.

