# OpenReview forum: "Short-PHD: Detecting Short LLM-generated Text with Topological Data Analysis After Off-topic Content Insertion"
_colmweb.org/COLM/2025/Conference — COLM 2025_

### Official Review · Reviewer_nqRY · 2025-05-10

**Rating:** 6
**Confidence:** 4
**Ethics Flag:** 1

**Summary:**

The paper proposes Short-PHD, a new zero-shot method for detecting short LLM-generated texts using Persistent Homology Dimension (PHD). It identifies a key limitation of the original PHD method (instability in estimating PHD for short texts due to “local density peaks”) and addresses this by inserting off-topic content (OCI) before PHD computation. The authors provide both theoretical justification and empirical evaluation, demonstrating that Short-PHD improves detection accuracy and robustness over prior zero-shot and supervised methods, especially for short texts.

**Reasons To Accept:**

**Novelty:** The idea of stabilizing PHD via off-topic content insertion is new and well-motivated.

**Empirical Evidence:** The paper provides experimental results on multiple datasets and LLMs, showing that Short-PHD generally outperforms compared baselines for short text detection. The sensitivity analyses on text length and the number of off-topic content pieces provide further support for the method's design choices.

**Robustness Evaluation:** The evaluation of Short-PHD against decoherence and paraphrasing attacks is a valuable contribution, suggesting the method’s potential practical utility in adversarial settings.

**Reasons To Reject:**

**Limited Theoretical Depth:** The mathematical justification for why OCI maintains the gap between human and machine PHD values is largely empirical and heuristic rather than formal.

**Motivation for Off-Topic Content Choice:** While the paper states that the off-topic content is generally unrelated and provides the specific content used, a more detailed justification for the choice of this particular off-topic content, and a discussion of whether the nature of the off-topic content impacts the results, would strengthen the method’s explanation. Furthermore, repeated insertion of the same set of fixed contents may create patterns detectable by adversaries or confound downstream NLP tasks.

**Writing Quality:** The overall writing is understandable, but there are opportunities for improvement in clarity, conciseness, and flow. For example, making the main body more self-contained.

---

> ### Author Response · Authors · 2025-05-30
>
> The authors thank Reviewer nqRY very much for the review. We address your concerns as follows.
>
> ```
> [Q1] Limited Theoretical Depth: The mathematical justification for why OCI maintains the gap between human and machine PHD values is largely empirical and heuristic rather than formal.
> ```
>
> We acknowledge that OCI is more of an empirical finding than a theoretical one. Inspired by previous work [1], we find that the PHD method encounters a phenomenon named local density peaks, which are particularly problematic for short texts, as they lack sufficient variability to sample out of the peak. Therefore, we provide a simple heuristic (OCI) to solve the problem.
>
> In the method section, we start by positing some hypotheses and then verify them empirically, which is similar to other zero-shot methods, such as DetectGPT [2] and Fast-DetectGPT [3]. We believe that these hypotheses are well-justified and empirically verified. To this end, we build our Short-PHD method.
>
> ```
> [Q2] Motivation for Off-Topic Content Choice: While the paper states that the off-topic content is generally unrelated and provides the specific content used, a more detailed justification for the choice of this particular off-topic content, and a discussion of whether the nature of the off-topic content impacts the results, would strengthen the method's explanation. Furthermore, repeated insertion of the same set of fixed contents may create patterns detectable by adversaries or confound downstream NLP tasks.
> ```
>
> Thank you for your comment. We further verify the importance of the off-topic content selection as follows. Specifically, we examine the impact of on-topic insertion. We modify Short-PHD with a simple heuristic by inserting the same content as the detected text. This ensures that the inserted content is on the same topic as the detected text. The results are shown below.
>
> |Domain|Wikihow|||||Eli5|||||
> |-|:-:|:-:|:-:|:-:|:-:|:-:|:-:|:-:|:-:|:-:|
> ||OPT|OPT-iml|NeoX|GPT-3.5|GPT-4o|OPT|OPT-iml|NeoX|GPT-3.5|GPT-4o|
> |Short-PHD (on-topic)|0.66|0.67|0.73|0.60|0.60|0.60|0.60|0.60|0.56|0.55|
> |Short-PHD (off-topic)|**0.78**|**0.73**|**0.77**|**0.80**|**0.79**|**0.84**|**0.82**|**0.82**|**0.87**|**0.86**|
>
> The results demonstrate that inserting off-topic content yields higher detection scores across all models and domains compared to inserting the same detected text. Simply increasing the length of the detected text does not improve detection. Therefore, it is crucial to ensure that the inserted text is off-topic.
>
> For your last point regarding the security concern, we also acknowledge that inserting the same content may introduce vulnerabilities to some potential detection attacks. Specifically, the uniformity of the inserted content could be exploited by adversaries, thereby weakening the robustness of the model by some prompt-based attacks. We leave it as a valuable future work.
>
> ```
> [Q3] Writing Quality: The overall writing is understandable, but there are opportunities for improvement in clarity, conciseness, and flow. For example, making the main body more self-contained.
> ```
>
> Thank you for your valuable comments regarding the writing quality. We will revise the manuscript to improve the quality.
>
> [1] Tulchinskii, E., Kuznetsov, K., Kushnareva, L., Cherniavskii, D., Nikolenko, S., Burnaev, E., ... & Piontkovskaya, I. (2023). Intrinsic dimension estimation for robust detection of ai-generated texts. Advances in Neural Information Processing Systems, 36, 39257-39276.
>
> [2] Mitchell, E., Lee, Y., Khazatsky, A., Manning, C. D., & Finn, C. (2023, July). Detectgpt: Zero-shot machine-generated text detection using probability curvature. In International Conference on Machine Learning (pp. 24950-24962). PMLR.
>
> [3] Bao, G., Zhao, Y., Teng, Z., Yang, L., & Zhang, Y. Fast-DetectGPT: Efficient Zero-Shot Detection of Machine-Generated Text via Conditional Probability Curvature. In The Twelfth International Conference on Learning Representations.

---

### Official Review · Reviewer_hvbd · 2025-05-13

**Rating:** 7
**Confidence:** 3
**Ethics Flag:** 1

**Summary:**

Tulchinskii, et al. 2024 introduced a method for detecting LLM-generated text based on the persistent homology dimension (PHD) of contextual embeddings. This paper introduces an extremely simple modification to that method, showing that detection performance can be substantially improved by simply appending a paragraph of text before the input, especially for shorter input texts. The paper argues that this approach works because PHD estimators are unstable given only a small number of inputs, leading to high variance. Adding dummy text before the input results in a larger number of vectors to be input to the PHD estimator, which stabilizes its result.

The paper also provides comparisons to a number of standard zero-shot baselines, and also provides some ablations on paraphrasing and decoherence attacks

**Questions To Authors:**

1. For figure 4(d), did you retrain the RoBERTa model on LLaMA? My guess is probably not, and the section on sensitivity to white-box detection should be updated to reflect that if so

**Reasons To Accept:**

1. The problem of detecting LLM-generated text is an important one. The method in this paper is extremely simple to implement and provides clear gains over a well-known baseline (PHD).

2. The method in this paper provides large improvements to the detection of very short texts, which is an important but understudied problem in the LLM detection space. However, it is worth noting that high-accuracy detection may not be possible for short texts [1].

[1] Chakraborty, S., Bedi, A. S., Zhu, S., An, B., Manocha, D., & Huang, F. (2023). On the possibilities of ai-generated text detection. arXiv preprint arXiv:2304.04736.

**Reasons To Reject:**

1. My main concern is that the baselines used in this paper are excessively weak, so while the method is clearly better than PHD, it's hard to determine how useful it is in real-world settings. In particular, DetectGPT with mismatched scoring and target models is a very weak method (cf. Figure 6 of the DetectGPT paper), and the RoBERTa-based classifier is trained on the outputs of GPT-2, which is substantially weaker than the target models. I'd consider implementing some stronger baselines, e.g., those in [2] or [3]. Of course, different methods have different tradeoffs, but I would recommend clearly delineating those when comparing models

2. Nitpicking here, but the paper makes the claim that "since PHD measures the connectedness of embeddings, limiting the inserted content to off-topic (i.e., irrelevant) content ensures that the PHD increases for all inputs because text embeddings are less connected," but I'm not sure this claim is actually supported by the results. Do you have any evidence that the inserted text needs to be off-topic? Or will any prefix text do just as well?

[2] Hans, A., Schwarzschild, A., Cherepanova, V., Kazemi, H., Saha, A., Goldblum, M., ... & Goldstein, T. (2024). Spotting LLMs with binoculars: Zero-shot detection of machine-generated text. arXiv preprint arXiv:2401.12070. [3] Verma, V., Fleisig, E., Tomlin, N., & Klein, D. (2024, June). Ghostbuster: Detecting Text Ghostwritten by Large Language Models. In Proceedings of the 2024 Conference of the North American Chapter of the Association for Computational Linguistics: Human Language Technologies (Volume 1: Long Papers) (pp. 1702-1717).

---

> ### Author Response · Authors · 2025-05-30
>
> The authors thank Reviewer hvbd very much for the review. We detail our responses for each of your comments as follows.
>
> ```
> [Q1] My main concern is that the baselines used in this paper are excessively weak, so while the method is clearly better than PHD, it's hard to determine how useful it is in real-world settings. In particular, DetectGPT with mismatched scoring and target models is a very weak method (cf. Figure 6 of the DetectGPT paper), and the RoBERTa-based classifier is trained on the outputs of GPT-2, which is substantially weaker than the target models. I'd consider implementing some stronger baselines, e.g., those in [2] or [3]. Of course, different methods have different tradeoffs, but I would recommend clearly delineating those when comparing models
> ```
>
> Thank you for your valuable suggestion, we implemented two more recent baselines, which are Fast-DetectGPT [1] and DNA-GPT [2], in our comparison. We conducted experiments on black-box detection in the tables below for texts with 50 tokens each. The setting is kept the same as in the previous experiment in Table 4.
>
> |Domain|Wikihow|||||Eli5|||||
> |-|:-:|:-:|:-:|:-:|:-:|:-:|:-:|:-:|:-:|:-:|
> ||OPT|OPT-iml|NeoX|GPT-3.5|GPT-4o|OPT|OPT-iml|NeoX|GPT-3.5|GPT-4o|
> |Fast-DetectGPT|0.73|**0.74**|0.74|0.77|0.76|0.82|0.81|0.80|0.83|0.84|
> |DNA-GPT|0.57|0.55|0.58|0.64|0.64|0.55|0.50|0.55|0.73|0.74|
> |Short-PHD|**0.78**|0.73|**0.77**|**0.80**|**0.79**|**0.84**|**0.82**|**0.82**|**0.87**|**0.86**|
>
> As depicted, Short-PHD consistently outperforms DNA-GPT across different models and domains. Short-PHD also achieves an overall improvement in performance compared to Fast-DetectGPT for short text detection. The experimental results highlight the effectiveness of Short-PHD.
>
> ```
> [Q2] Nitpicking here, but the paper makes the claim that "since PHD measures the connectedness of embeddings, limiting the inserted content to off-topic (i.e., irrelevant) content ensures that the PHD increases for all inputs because text embeddings are less connected," but I'm not sure this claim is actually supported by the results. Do you have any evidence that the inserted text needs to be off-topic? Or will any prefix text do just as well?
> ```
>
> Thank you for your comment. It is an insightful comment on where we need off-topic content rather than any content. To answer this question, we conduct an empirical validation through a simple heuristic by inserting the same content as the detected text. This ensures that the inserted content is on the same topic as the detected text. The results are shown below.
>
> |Domain|Wikihow|||||Eli5|||||
> |-|:-:|:-:|:-:|:-:|:-:|:-:|:-:|:-:|:-:|:-:|
> ||OPT|OPT-iml|NeoX|GPT-3.5|GPT-4o|OPT|OPT-iml|NeoX|GPT-3.5|GPT-4o|
> |Short-PHD (on-topic)|0.66|0.67|0.73|0.60|0.60|0.60|0.60|0.60|0.56|0.55|
> |Short-PHD (off-topic)|**0.78**|**0.73**|**0.77**|**0.80**|**0.79**|**0.84**|**0.82**|**0.82**|**0.87**|**0.86**|
>
> The results demonstrate that inserting off-topic content yields higher detection scores across all generators and domains compared to inserting the same detected text. It provides some initial evidence that simply increasing the length of the detected text does not improve detection.
>
> ```
> [Q3] For figure 4(d), did you retrain the RoBERTa model on LLaMA? My guess is probably not, and the section on sensitivity to white-box detection should be updated to reflect that if so.
> ```
>
> Thank you for your comment. We indeed use the OpenAI pretrained RoBERTa model and do not fine-tune it [3]. We will incorporate this clarification as you suggested in the updated version.
>
> [1] Bao, G., Zhao, Y., Teng, Z., Yang, L., & Zhang, Y. Fast-DetectGPT: Efficient Zero-Shot Detection of Machine-Generated Text via Conditional Probability Curvature. In The Twelfth International Conference on Learning Representations.
>
> [2] Yang, X., Cheng, W., Wu, Y., Petzold, L. R., Wang, W. Y., & Chen, H. DNA-GPT: Divergent N-Gram Analysis for Training-Free Detection of GPT-Generated Text. In The Twelfth International Conference on Learning Representations.
>
> [3] Solaiman, I., Brundage, M., Clark, J., Askell, A., Herbert-Voss, A., Wu, J., ... & Wang, J. (2019). Release strategies and the social impacts of language models. arXiv preprint arXiv:1908.09203.

---

> > ### Comment · Reviewer_hvbd · 2025-06-04
> >
> > Thanks for the thorough response! I raised my score from 6 -> 7.
> >
> > In particular, I think the on-topic vs. off-topic results are quite striking, and this fully addresses my concern #2. I appreciate the inclusion of stronger baselines too, although the method's improvements over Fast-DetectGPT are relatively small, so it would be worth running a statistical test to verify the model actually outperforms this baseline.

---

> > > ### Author Response · Authors · 2025-06-06
> > >
> > > Thank you for the thoughtful review and for raising the score. We sincerely appreciate your feedback.

---

### Official Review · Reviewer_9J1C · 2025-05-13

**Rating:** 6
**Confidence:** 3
**Ethics Flag:** 1

**Summary:**

This paper addresses the challenge of detecting short LLM-generated texts by proposing Short-PHD, an extension of the Persistent Homology Dimension (PHD) method. The major innovation is the off-topic content insertion (OCI) technology, which stabilizes PHD estimation for short texts by prepending irrelevant content.

The authors claimed that OCI mitigates local density peaks in PHD estimation, thus improving detection robustness. Experiments on public and generated datasets show that Short-PHD outperforms zero-shot baselines (e.g., PHD, DetectGPT) and even supervised methods like RoBERTa, achieving a 25–30% improvement in AUC for 50-token texts.

**Questions To Authors:**

What is the time and storage complexities of Short-PHD (and PHD), and how does it compare to likelihood-based methods that requires one-time or multiple times of LLM inference.

**Reasons To Accept:**

Novelty: The OCI technique is creative and well-motivated, addressing a clear gap in PHD-based detection for short texts.

Technical soundness: The method is grounded in topological data analysis, with rigorous mathematical equations,  and a clear algorithmic workflow (Algorithm 1). The two hypotheses for OCI and Short-PHD are also empirically validated.

Good performance: Short-PHD achieves significant improvements over PHD and some other baselines, and maintains robustness across domains and against the adversarial paraphrasing attacks.

The paper is well-structured.

**Reasons To Reject:**

There could be a broader coverage of baseline methods.

1. Although Fast-DetectGPT (Bao et al, 2024) is cited in addition to DetectGPT (Mitchell et al., 2023), its performance on your dataset is not  added to Table 4. According to the original Fast-DetectGPT paper, it should have better overall detection acc for most cases. Just wondering why it is not reported.

2. Here are some recent detection methods that may be considered as a comparison detection:
- FourierGPT, Xu et al., 2024: A psycholinguistics-inspired method that use spectral features of surprisal for detection, which works particularly well on short text (out-performing Fast-DetectGPT on PubMed)
- GPT-who, Venkatraman et al., 2024: Also a psycholinguistics-inspired method that utilizes UID (uniform information density) features, which has also out-performed DetectGPT on some datasets.
- Han et al., 2024, Spotting LLMs With Binoculars: Zero-Shot Detection of Machine-Generated Text
- Yang et al., 2023, DNA-GPT, which provides more interpretability at n-gram levels.

---

> ### Author Response · Authors · 2025-05-30
>
> The authors thank Reviewer 9J1C very much for the review. We detail our responses for each of your comments as follows.
>
> ```
> [Q1] There could be a broader coverage of baseline methods.
>
> Although Fast-DetectGPT (Bao et al, 2024) is cited in addition to DetectGPT (Mitchell et al., 2023), its performance on your dataset is not added to Table 4. According to the original Fast-DetectGPT paper, it should have better overall detection acc for most cases. Just wondering why it is not reported.
> ```
>
> Thank you for your suggestions on providing the latest baseline methods. We compared two more recent baselines, which are Fast-DetectGPT [1] and DNA-GPT [2] according to your comment. We conducted experiments on black-box detection in the tables below for texts with 50 tokens each. The setting is kept the same as in the previous experiment in Table 4.
>
> |Domain|Wikihow|||||Eli5|||||
> |-|:-:|:-:|:-:|:-:|:-:|:-:|:-:|:-:|:-:|:-:|
> ||OPT|OPT-iml|NeoX|GPT-3.5|GPT-4o|OPT|OPT-iml|NeoX|GPT-3.5|GPT-4o|
> |Fast-DetectGPT|0.73|**0.74**|0.74|0.77|0.76|0.82|0.81|0.80|0.83|0.84|
> |DNA-GPT|0.57|0.55|0.58|0.64|0.64|0.55|0.50|0.55|0.73|0.74|
> |Short-PHD|**0.78**|0.73|**0.77**|**0.80**|**0.79**|**0.84**|**0.82**|**0.82**|**0.87**|**0.86**|
>
> As shown in the table, Short-PHD consistently outperforms or matches the performance of the recent baselines. Specifically, our method surpasses DNA-GPT significantly across different models and domains. Short-PHD also achieves an overall improvement in performance compared to Fast-DetectGPT for short text detection. These findings highlight the effectiveness of Short-PHD.
>
> ```
> [Q2] What is the time and storage complexities of Short-PHD (and PHD), and how does it compare to likelihood-based methods that requires one-time or multiple times of LLM inference.
> ```
>
> Thank you for your comment. The computation time of PHD and Short-PHD consists of two main parts: the embedding computation (on GPU) and the PHD computation (on CPU). It is hard to compute the time and storage complexities for both, but we report the empirical running time below. On our server with an H100 GPU and a 3.1GHz CPU (a moderate CPU), the embedding inference costs around 0.06s for texts with less than 100 tokens, and the PHD computation on a single core is around 0.49s. Note that it is easy to parallelize the PHD computation, but we only use one core. Compared to DetectGPT, the inference time is comparable, but PHD does not require the time-consuming perturbation process used in DetectGPT.
>
> [1] Bao, G., Zhao, Y., Teng, Z., Yang, L., & Zhang, Y. Fast-DetectGPT: Efficient Zero-Shot Detection of Machine-Generated Text via Conditional Probability Curvature. In The Twelfth International Conference on Learning Representations.
>
> [2] Yang, X., Cheng, W., Wu, Y., Petzold, L. R., Wang, W. Y., & Chen, H. DNA-GPT: Divergent N-Gram Analysis for Training-Free Detection of GPT-Generated Text. In The Twelfth International Conference on Learning Representations.

---

### Decision · Program_Chairs · 2025-07-08

**Decision:**

Accept

**Comment:**

The authors propose an extension of the persistent homology dimension (PHD) method for machine-generated text detection for the case of short texts. This is an important issue, as most detectors struggle with short texts. The newly proposed method Short-PHD has strong performance, even though it is an extremely simple modification of the original PHD method: add some extra off-topic text in order to stabilize PHD.

The additional results and the discussion are promising. Overall, this is good work, but the method is quite simple. I find this work borderline, leaning positive.

I am an expert, actively working on this topic, and I am very well familiar with related work.